# A Review of Village Ecosystem Structure and Stability: Implications for the Karst Desertification Control

**Li Lin, Kangning Xiong \*** , **Qi Wang, Rong Zhao and Jiayi Zhou**

School of Karst Science, State Engineering Technology Institute for Karst Desertfication Control, Guizhou Normal University, Guiyang 550001, China; 21010170534@gznu.edu.cn (L.L.); waldowangqi@gznu.edu.cn (Q.W.); zhaorong424@outlook.com (R.Z.); zhoujiayi2225@163.com (J.Z.)

\* Correspondence: xiongkn@gznu.edu.cn

**Abstract:** Villages are places where people gather and live. Due to economic and social development, the irrational behavior of the population has led to an imbalance in the village structure and has threatened the stability of the village ecosystem, resulting in the deterioration of the village environment. Therefore, it is of great necessity to study the structure and stability of village ecosystems and to optimize the structure of village ecosystems to better guide spatial planning and to restore village ecology. This study conducted a literature search and statistical analysis on the structure and stability of village ecosystems based on WOS and CNKI literature databases. We reviewed 105 relevant articles. The current research status and progress are clarified from structural characteristics, structural optimization, structure and function and stability study. To reveal the research achievements and deficiencies of research on the structure and stability of village ecosystems, the vital scientific issues that need to be addressed are summarized based on current research results. The study found that: (1) the quantity of studies on it were on the rise; (2) the study content mainly focused on structure and function (38%) and structural characteristics (21%); (3) the regions studied are mainly in Asia (73%), Europe (12%) and Africa (10%); and (4) research institutions are mainly colleges and universities. Therefore, future research should give attention to the following three aspects: strengthening the differentiation research on the spatio-temporal scale, qualitative and quantitative analysis of the influence of the Karst Desertification Control (KDC) village structure on stability; based on the mechanism of structure on function, appropriate village ecosystem structure should be established to improve ecosystem service function; based on the influence mechanism of structure on stability, the stability evaluation index system will be constructed so as to lay a solid foundation for the stability strategy of the KDC village ecosystem. By applying the strategy of structure optimization and stability improvement to the KDC village ecosystem, the service function of the Karst village ecosystem can be improved, which can provide scientific reference for the sustainable development of the KDC village ecosystem.

**Keywords:** village ecosystem; structure; stability; progress; Karst Desertification Control; literature review



## 1. Introduction

With the deterioration of natural environment and the interference of human activities, the ecological structure of the KDC village has changed. The village ecosystem is faced with pressure and challenges, such as the collapse and damage of dwellings, abandoned landscape nodes and insufficient service facilities, which cause harm to the normal life of residents. Therefore, reasonable planning is needed [1–3]. Some villages used the concept of village ecosystem structure optimization and ecological stability to alleviate the fragmentation of village landscape, to coordinate the relationship between village development and ecosystem and to realize the sustainability of village development [4,5]. The village ecosystem is composed of natural villages with certain distribution characteristics and scale [6], with production, living and ecological functions, which are embodied in carrying

population, providing products, improving the environment, providing ecological barrier and inheriting culture [7]. Village ecosystem structure includes all the physical properties and biological characteristics of the system [8,9], which is divided into external landscape structure and internal population, livelihood and energy structure. As an essential embodiment of structure of the KDC village ecosystem, a stable village landscape pattern can provide village residents with many ecosystem services (ES) that match their needs [10,11], solve village energy use and stabilize the village ecological balance [12]. The optimization of village landscape patterns can promote the intensive and efficient use of land resources and then adjust the PLE (production–living–ecology) function of the village ecosystem to stimulate the endogenous power of villages to cope with village decline, village ecosystem degradation and other problems [13,14]. Therefore, the perfection of structure can play a vital role in promoting regional economic development in KDC.

Serious human disturbance leads to the existence of KDC ecosystem landscape [15]. KDC villages are typical ecologically fragile areas [16,17]. High-intensity agricultural activities dominate these areas, and the Karst environment of villages has problems, such as vegetation degradation, severe "Karst drought", heavy soil and water loss and intensified Karst desertification. Activities result in deforestation by farmers, shortage of available water resources and reverse succession of biological communities [18,19]. Desertification is the ultimate result of land degradation in the Karst areas [20], and it is also one of the four major ecological problems in western China [21,22]. The fundamental problem of the ecological environment is that land use changes the ecosystem's structure [23]. Diversification of land resources provision is one of the guarantees for the security and effective supply of regional land resources. It is significant for areas with fragile ecological environments [24]. Therefore, the pluralism of the village ecosystem structure not only conforms to the stable characteristics of the system but is also the inevitable trend of sustainability of ecosystem.

The higher the connection intensity of the village ecosystem structure, the closer and more stable the connections between and within the villages; on the contrary, if there are isolated patches or corridors, the weaker the system's stability will be [25]. Structure and stability should be combined in the KDC to explore the correlation between them so as to better improve the supply capacity of ES [26]. A correct understanding of the stability of the village ecosystem structure can better control desertification in Karst areas while maintaining the normal functioning of the ecosystem. An increasing number of scholars have studied the stability of villages in conjunction with structure [27]. Some focus on the relationship between various structural factors in the village ecosystem [28], while others focus on the harmony and unity of the external landscape [29]. The stability of villages is discussed through the influence of structure on function, and the formation mechanism of structure stability is explained. However, hysteresis between the ecosystems structure and functions is particularly severe [30]. Therefore, it is necessary to study and monitor the ecosystem structure [31], to dynamically observe the impact of changes in ecosystem structure on the system and to make accurate and timely adjustments to keep the system in a relatively balanced and stable state.

The sustainable development of village ecosystems is of great importance to human and even global environment. The stability of village ecosystem structure has received increasing attention from scholars from all walks of life. However, there is no such research work on it. The current articles systematically search in order to better grasp the direction of development and to guide research on the stability of village ecosystem structure and development. By reviewing the research advances and signature results on the structure and stability of village ecosystems, we look forward to the key scientific and technical issues that need to be addressed in future research and point out the future directions for further research in KDC. It will provide scientific support for the stable development of KDC villages.

## 2. Materials and Methods

By systematically summarizing and organizing the scientific information obtained, we understand the development of the research field, compare and analyze it, and search for new research entry points around the research topic. Open up new research perspectives through this robustness and comprehensiveness of this approach [32]. By summarizing and comparing existing studies on the structure and stability of village ecosystems, key scientific and technical results as well as shortcomings are derived. Figure 1 illustrates the literature acquisition process.

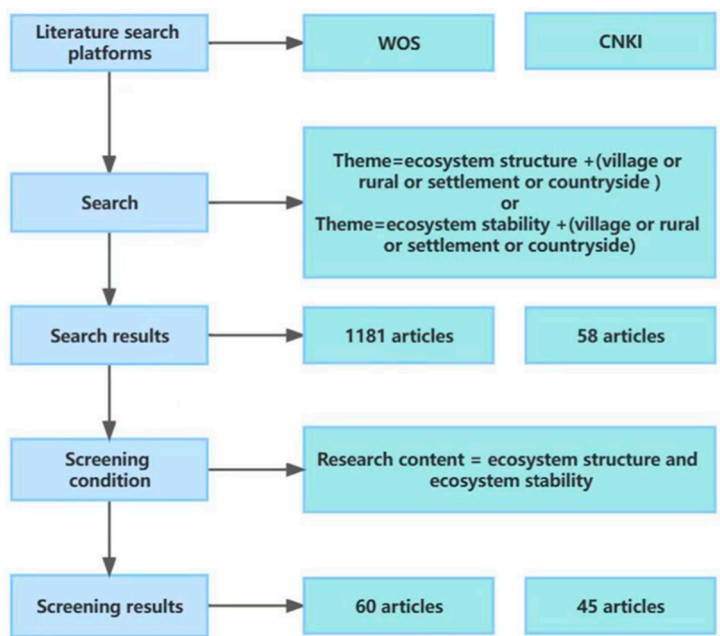

**Figure 1.** The literature acquisition process.

First, a literature search was carried out based on the core databases of China Knowledge Network (CNKI) and Web of Science (WOS). The deadline was 7 July 2022. Using "theme" as the search item in the WOS core database and taking "ecosystem structure" and "ecosystem stability" and "village" + "rural" + "settlement" + "countryside" as search terms, 1181 pieces of literature were found. In the CNKI, with "theme" as the search term, "ecosystem structure" and "ecosystem stability" and "village" + "settlement" + "village" + "countryside" + "village" as the search terms, a total of 58 pieces of literature were obtained.

Second, the screening criteria are systematically defined by the literature review procedure. The article's title, abstract and keywords were mainly reviewed: (1) The relationships between the components of the village structure are focused on and (2) the effect of structural changes on function and stability were given attention. A total of 1239 initially searched articles were manually screened. After screening titles, abstracts and keywords, full-text screening—which cannot identify a link with the topic—was performed. At least one of the remaining articles met the above criteria. Literature that had nothing to do with the structure and stability of village ecosystems was eliminated, leaving 105 papers. Among them, there are 45 Chinese literature, including 17 master's degree theses, 4 doctoral degree theses, 24 journal papers, 0 conference papers, 0 newspaper papers, 0 books, 0 achievements, 0 yearbooks, 0 patents and 0 standards. The remaining 60 are foreign literature.

## 3. Results

In an effort to more intuitively comprehend and explore the influence of village ecosystem structure on its stability, a statistical analysis of 105 articles was conducted to explore the study advances and signature results.

### 3.1. Annual Publication Volume of Literature

The number of published articles is rising (Figure 2); we summarize the trend in the following phases. From 1989 to 2010, the first stage was the embryonic stage. Literature began to appear one after another, and the annual publication volume of literature was less than 5. In the second stage, the number of articles showed fluctuating growth from 2011 to 2016. The last phase, from 1989 to 2010, showed a steady trend of growth, with an average annual number of 10 or more publications.

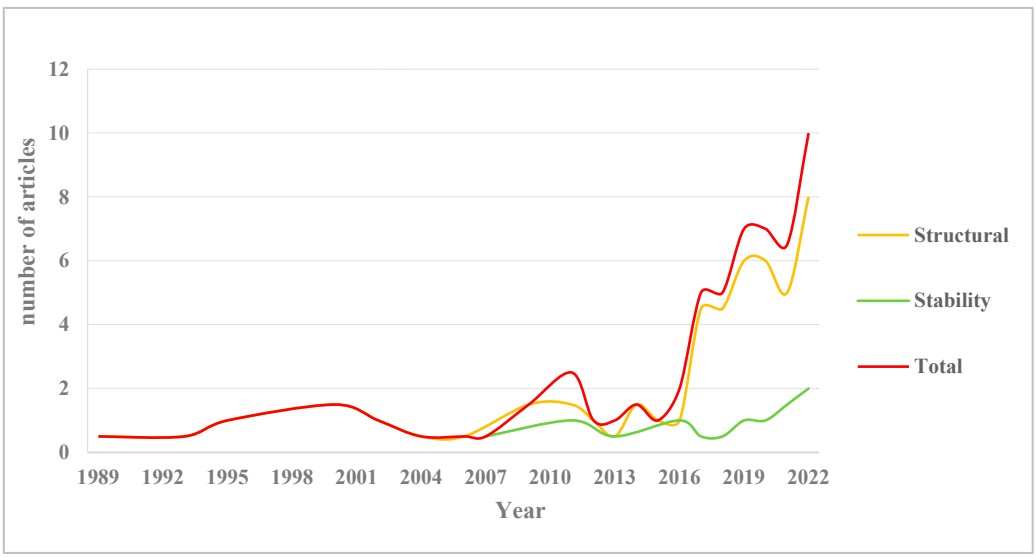

**Figure 2.** Annual publication of literature.

### 3.2. Classification of Literature Research Content

All the literature screened was classified and summarized around the main studies into structural characteristics, structural optimization, structure and function, stability studies and other related studies. Structure and function accounted for 38%; structural features accounted for 21%; structural optimization accounted for 16%; stability studies accounted for 18%; and the last 7% were other types. In proportion to the type of literature, studies of village ecosystems have focused on structure, function and structural characteristics, and stability studies are still in the exploratory development stage. Therefore, in the village ecosystem, the research on stability should be strengthened in order to clarify the relevant connotation of ecosystem stability.

### 3.3. Regional Distribution of Literature Research

The development of village ecosystems also has regional characteristics due to the differences in various natural and human factors in different regions. According to the collation of literature, the literature research field of the obtained literature was statistically analyzed. In terms of the volume of published literature, the scientific literature research on the structure and stability of village ecosystems covers 32 countries, except for a few articles published in national official languages and local journals that have not been searched. Most studies focused on Asia (73%), Europe (12%) and Africa (10%), with China having the largest number and share of 75 articles. Italy and the United States followed with three articles. It was followed by three in Spain and three in Russia. The number of literature from other countries/regions is only one.

### 3.4. Distribution of Authors' Research Units

Based on the affiliated institutions of the first author, this paper analyzes the publication units related to the content of the study. The analysis revealed that research units from 15 countries published relevant articles, with China (71%) leading the number of

articles, followed by the United States (8%), Italy (4%) and Spain (3%). The author's affiliations were divided into two categories: universities and research institutes (centers), which account for 79% and 20%, respectively, and the committee, which accounts for 1%. The relevant units are mainly universities: There are Northwest Agriculture and Forestry University, Chongqing University, Beijing Forestry University, Beijing Normal University, Guilin University of Technology, etc.

## 4. Research Advances and Landmark Achievements

### 4.1. Structural Characteristics

The village ecosystem was analyzed from the perspectives of Karst natural ecological environment and population and economic status, and the structure of the village ecosystem was found to include internal and external structures, laying a realistic foundation for clarifying the connotation of village structure.

Due to the fragile environment of the Karst village ecosystem, thin soil layer, a little cultivated land, tension between man and land and blocked traffic, the development of agriculture and village economy is relatively backward [33]. The ecosystem is affected by external environmental conditions and internal elemental organization, and its structure is broken and diverse, complex and changeable [34]. The village ecosystem includes not only the landscape pattern of the village but also the internal population structure, livelihood structure and energy structure, which all affect the regular operation of the ecosystem. The external structure is mainly composed of the village landscape, and the four main elements of the spatial structure of the landscape ecosystem are spots, corridors, bases and edges [35]. Most previous studies focused on the analysis of landscape pattern but ignored the significant role of internal structure in the system [36–38]. Changes in ecosystem structure affect the land structure of PLE of village, ES and farmers' livelihoods [39–43]. The individual characteristics of farmers are mainly reflected in the living capital and family structure [44]. And in the operation of the system, they will react on landscape pattern. That is, the internal and external structures of the ecosystem interact and influence each other. Variations in population, energy and livelihood structure within villages will change the landscape structure of villages, which in turn will affect the farming area and assets of the villagers and ultimately change the livelihood decisions of the villagers. Farmer cooperatives are one way in which the interests of farmers can be promoted, and the increase or decrease of assets changes the livelihood decisions of villagers [45].

As a result of poor regional environmental conditions and backward economic development, the young talent go out to make a living and improve their families' economic ability to resist risks [46,47]. At the same time, population outflows have led to village decline and increased risks to the lives and health of vulnerable groups in villages [48–50]. Therefore, comprehensive consideration of the internal and external structure of the village is a necessary prerequisite for promoting the sustainable development of the KDC village ecosystem.

### 4.2. Structural Optimization

4.2.1. The Village Ecosystem Structure Has Been Adjusted through Methods Such as Increasing the Heterogeneity of Landscape Patterns and the Diversity of Livelihood Structures, Which Are Necessary Prerequisites for Optimizing the Structure of Karst Village Ecosystems

The optimization of the structure of the village ecosystem is based on the idea of system theory so that the structures within the village ecosystem can form the best-quantified ratio relationship in terms of material, information, capital exchange and energy flow, according to the specific conditions of the village so that the system can achieve certain ecological benefits [51]. With the development of village studies, the landscape has attracted extensive attention, and much literature is concerned with landscape patterns. Landscape pattern is the specific manifestation of landscape heterogeneity. Through regulating external energy input, landscape patterns can be changed to make them more suitable for human survival [52]. Village landscape is not only a landscape type [53] but also a combination of human disturbance characteristics and natural ecological structure established on the basis

of natural landscape [54]. The landscape is a spatial-heterogeneous area [53], with patch size, straight corridors, high or low connectivity and more diverse substrates, which together constitute a rich and diverse landscape pattern (Figure 3) [55,56]. Heterogeneity is closely related to anti-interference ability, resilience, system stability and biodiversity [57,58]. The stronger the heterogeneity is, the better the biodiversity is, the more stable the landscape structure is and the stronger the resistance to external disturbance is. On the contrary, the rationality of the landscape pattern is gradually lost, and the homogeneity will cause the spread of disturbance [59]. Structural adjustment should be implemented to increase the heterogeneity of landscape patterns, enrich the diversity of village landscape patterns, increase its anti-interference ability and ensure the stability of the village ecosystem.

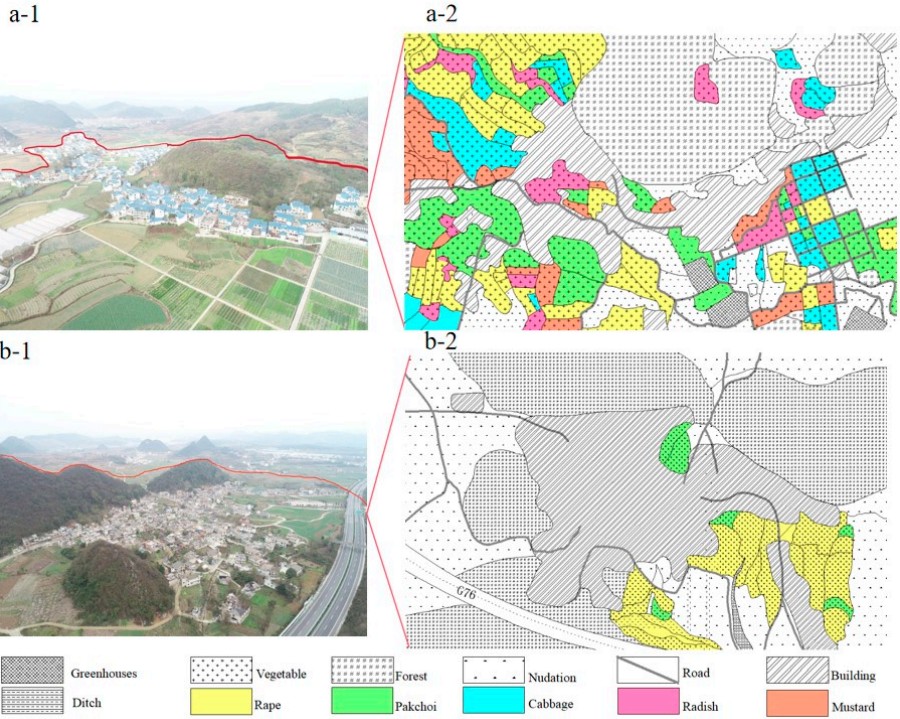

**Figure 3.** Landscape pattern of village ecosystem. (**a-1**) Youqi Village—Photo taken by drone. (**a-2**) Youqi Village—Landscape pattern diagram of village ecosystem. (**b-1**) Wangjiazhai Village—Photo taken by drone. (**a-2**) Wangjiazhai Village—Landscape pattern diagram of village ecosystem.

In addition, as corridors in village landscape, the optimization degree of rivers and roads is closely related to the connectivity of village ecosystem. Therefore, carrying out ecological restoration of waters and road improvement work to optimize the layout of water systems and roads is an important prerequisite for structural optimization. Optimize the layout of villages, rationalize land planning, improve the stability of villages, and decrease haphazard developers [36,60,61]. The structure of village ecosystem will inevitably involve the interaction of the man–land relationship. Livelihood structure, energy structure, population structure and landscape structure jointly act on the village ecosystem. However, at present, there are few optimization schemes involving internal and external structures, which need to be further explored.

### 4.2.2. By Constructing a Dynamic Monitoring System, the Structural Change Characteristics of the Village Ecosystem under Different Spatial and Temporal States Were Revealed, Which Is of Practical Significance for the Optimization of Village Ecosystem Structure in Karst Areas

One aspect of ecosystems that is rarely addressed is the effect of temporal and spatial heterogeneity on the system [62].

Different stages of village ecosystem structure with the limitations of time and need for different periods of the structure for dynamic monitoring. When the landscape pattern of the village cluster presents slight dynamic changes, there may be significant hidden ecological hazards in the local settlements [31]. The possible time lag between human disturbance and the subsequent response of the ecosystem, i.e., the difference in time between the effects of changes in ecosystem structure on the ecosystem. Therefore, the time trajectory of ecosystem structure and function should be taken into account when making ecological restoration plans [30].

Static evaluation is an assessment of the structural status of an ecosystem for a specific time [16], while village ecosystems located in different regions have heterogeneous structure and characteristics. Different ecosystem structures will pose disparate threats to the physical stability, spatial continuity and anti-interference ability of the landscape, so it is necessary to compare different village ecosystems. In a variety of geographical conditions, villages present different landscape structures, comparing the mountain village structure with the water network, analyzing various development directions of structure and providing a theoretical basis for the village development [63]. Therefore, dynamic monitoring of different village ecosystem structures is needed. It is the basis of long-term stability and sustainable development to optimize and improve the structure and to realize the beautiful village ecological environment [59].

### 4.3. Structure and Function

Through the Analysis of the Nature of Village Ecosystems, the Mechanism of the Role of System Structure on Function Is Clarified, Providing a Theoretical Basis for the Function Enhancement and Structural Optimization of Village Ecosystems

Function refers to the nature and efficiency of system and external environment in the interaction process [64]. Structure and function interact with each other, and villages have economic, social and ecological functions that cannot be replaced by cities [65]. However, nowadays, there are many problems in the village ecosystem, which are mainly manifested as lack of supporting facilities for public services in living space, inefficient and fragmented production space and degradation of ecological service functions in ecological space [26]. Landscape service is the one of the most appreciated services [66]. Scientific classification of ecosystem functions and structures is an important prerequisite for improving village land use efficiency and optimizing landscape pattern allocation. Due to the lack of planning guidance and constraints for a long time, the village ecosystem is facing problems, such as chaotic spatial development order, generally small and scattered village layout scale and low efficiency of land resource utilization [67], deterioration of human settlement environment and insufficient endogenous power of village development [68]. The land has the property of bearing multiple functions, and the functions of PLE are the most basic of landscape. The dynamic process of interaction, game and coupling of the PLE functions leads to a more prominent role of the lead function of land use in system [69]. Village function is a characteristic of beneficial influence produced by a specific regional system in a certain stage of development through its own potential [70]. Optimizing the village ecosystem's structure and strengthening its function are vital content and foothold of ecological agriculture construction in China [51]. Based on the background conditions of natural resources in villages, planning the layout of the three functional areas according to local circumstances and realizing the coordinated development of each functional area has become an essential issue to be solved in order to build a harmonious coexistence between human beings and nature, implement the rural rejuvenation policy and promote the sustainable development of villages [71,72].

*4.4. Stability Study*

4.4.1. By Understanding the Connotation of Village Ecological Stability and Clarifying the Characteristics of Village Ecosystem Stability in Karst Areas, It Is an Important Foundation for Improving the Stability of Village Ecosystems

The general term "stability" is vague [73], and pure stability has no practical significance in ecology [74]. The physical meaning of stability is that the state of a system remains constant over a certain time interval [75]. Therefore, the connotation of stability is understood as: The system can adapt to the change of environmental factors and can maintain its own life state [76]. Landscape stability refers to the ability of landscape to remain unchanged, which can be divided into two categories: resistance stability and resilience stability [77], namely the resistance to disturbance and the resilience after disturbance [36]. Diversified livelihood structure means that farmers give full play to their subjective initiative and maximize the combination of agricultural production time and livelihood sources to ensure maximum household income. At the demographic level, this is mainly achieved by young people working outside the home and middle-aged and elderly people adopting diversified lifestyles, thus improving the stability of life and demographic structure. In the KDC villages that are not suitable for large-scale agricultural production, the relationship between land, nature, people and farming time should be properly handled, and the "hard-working economy" is the fundamental reason for the long-term survival and lasting prosperity of villagers [78]. A correct understanding of the definition of stability is a rather vital basis for the study of village ecosystem [79,80]. Reducing ecosystem vulnerability and increasing ecosystem resilience and stability is critical [81].

4.4.2. By Analyzing the Corresponding Relationships between Villages and Their Surroundings, the Factors Influencing the Stability between Villages Are Clarified, Which Can Contribute to the Improvement of the Stability of Village Boundary Ecosystems

Each village is composed of its own unique structure, which is formed in a specific geographical environment and social background. It is not existent in isolation but is closely related to various surrounding villages or environments [82]. Changes in slope and elevation of natural factors may affect the distribution and construction of village settlements. In addition, due to the different land policies implemented in each village, there are large differences in the stability and internal stability at the junction of the ecosystems of two adjacent villages; therefore, there is an urgent need to strengthen the research on the factors influencing stability between villages [83,84]. The species and species richness in the fringe of patches and substrates are different from those in the interior. The wider the fringe is, the more beneficial it is to protect the internal ecosystem [53]. The complexity and stability of the spatial structure reflect the characteristics of the village patches. The complexity of village patches is the basis for the evolution of villages, while the stability of patches represents the future development trend [61]. Clear boundary concepts can promote village stability. The tangible boundary of villages focuses on the significance of geographical location and administrative division, while the intangible focuses on the significance of culture and economy [85]. The stability of the village ecosystem must be maintained through material circulation, energy flow and information transmission with other ecosystems. Considering the stability maintenance mechanism of the village boundary ecosystem can effectively avoid the fragmentation of the landscape pattern and then form a stable village ecosystem, maintaining a stable ecological balance between the village ecosystem and the external circumstances and, finally, achieving village stability [29].

4.4.3. By Analyzing the Mechanism of the Role of Ecosystem Structure on Stability, Exploring the Mechanism of the Role of Both Is the Theoretical Basis for Proposing Strategies to Enhance Ecosystem Stability

The characteristics of the structure of villages have obvious irregularity, instability and lack of uniformity [86,87]. At present, a unified strategy to improve the stability of the village ecosystem has yet to be formed. The ecological surroundings in Karst regions is fragile. The Karst desertification caused by ground fragmentation and soil erosion is

essentially the degradation or disappearance of ecosystem function caused by the destruction of ecosystem structure [88,89]. Its ecosystem has strong sensitivity and poor stability, which brings difficulties to the rational development and utilization of the region [90,91]. Building a village ecosystem with harmonious spatial structure and ecological stability is necessary. In-depth diagnosis of affecting elements of ecosystem stability in Karst areas and the establishment of a structural model for the evaluation of village ecosystem stability will help to propose strategies for improving ecosystem stability. It is of great significance to reasonably regulate the stability of the village ecosystem and better construct the Karst ecological environment [92]. Firstly, the village landscape configuration should be optimized at a high level, and then the specific land structure and ratio should be optimized. The resulting land planning can establish a stable ecological environment for the rural biological community in a region, which is favorable to village stability [29,93]. According to the structural characteristics of village ecosystem with different levels of rock desertification, the layout of cultivated land is adjusted to implement diversified livelihood strategy structure [94–97]. Develop bio-energy to increase the proportion of farmers' living energy, improve economic benefits, ecological benefits and quality of life [98]. Analyzing the courtyard structure can optimize the structure and improve the stability [99].

## 5. Discussion

### 5.1. Key Scientific Issues That Urgently Need to Be Addressed

5.1.1. In View of the Unclear Factors Affecting the Stability of the Ecosystem at the Junction of Two Adjacent Villages, the Research Should Be Strengthened

The ecological situation at the junction of two adjacent village ecosystems is often easily ignored. The more complex the land structure is caused by different land management measures, the greater the probability of ecological risk at the village boundary, and its stability will decrease accordingly [31,61]. Other researchers have shown that there are different species and species richness at the edges of patches and substrates at ecosystem junctions than in the interior. The wider the edge zone, the more beneficial it is to protect the internal ecosystem, and the stronger the anti-risk ability of the system is, in other words, the stronger the stability of the ecosystem [53]. In the unregulated village boundaries, the more complex the ecosystem structure, the greater the landscape fragmentation degree and the lower the boundary stability, but the stronger the stability of the ecosystem within each village (Figure 4). Therefore, the study of factors influencing the stability of adjacent villages is strengthened to enhance the stability of the junction of village ecosystems.

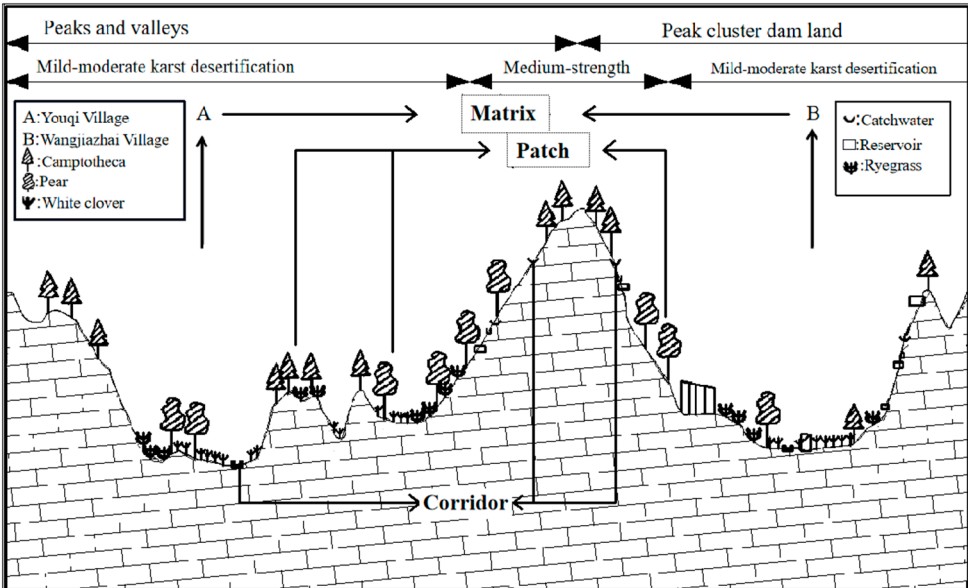

**Figure 4.** Landscape pattern profile at the junction of two adjacent village ecosystems.

### 5.1.2. In View of the Lack of Research on the Combination of Ecosystem Structure and Stability in KDC Village, a Research Framework Suitable for It Should Be Established

At present, most of the studies on ecosystem focus on structure and function [100,101], structural change [102,103] and other aspects, and all of them focus on a single direction, and there is a lack of research on the combination of structure and stability of KDC village ecosystem. In addition, in the previous structural studies, only one aspect of external landscape or internal livelihood, population and energy was involved, and the overall structure of villages was not completely discussed, which had certain limitations [104,105]. Therefore, the coupling relationship between internal and external structure and stability should be explored in future studies. From the landscape pattern and livelihood level, elucidating the relationship between stability and the structures that conclude the tangible and intangible of KDC village ecosystems, and revealing mechanisms for improving structure-based system stability. To explore how structural changes affect the stability of village ecosystem [106,107]. Landscape pattern and livelihood structure are natural and human structures in the structure of village ecosystem, and the interaction between them and the stability of village ecosystem can be discussed to facilitate the optimization and stability improvement of village ecosystem structure itself. To study the village ecosystem in KDC, we should consider many factors comprehensively and make clear the internal relationship between structure and stability.

### 5.1.3. In View of the Village Ecosystem Function and Stability Research Does Not Consider the Space and Time Differentiation, Should Strengthen That on time Scales

There are many studies on the status quo of village function and village function types [108–110]. However, it only stays in the dominant function, and more stays in the structural representation of the village. Lack of systematic and in-depth research on the source and cause of the diversity of village functions [111–113]. Ignore the time phase of evolution research, less to measure of the differences of village ecosystems. There is no summary analysis of the performance of the same function in different villages, so it is difficult to explain the source of village function gap [114]. The drivers of ecosystem stability in KDC villages are different at different times and the degree of stability varies [115,116]. Only through the dynamic evaluation of the function of village ecological structures in different periods can we gain a deeper understanding of the key drivers of stability change at each stage [117–119]. It points out the direction for the study of village ecosystem in KDC.

### 5.1.4. In View of the Unclear Selection Criteria of Evaluation Indexes for the Stability of Village Ecosystem, the Evaluation Index Should Be Constructed

Stability refers to the resistance of the system to interference and its recovery ability after interference. In KDC areas, due to the disparate natural factors and degree of human disturbance, the villages in the region are also located in different landform types and Karst desertification levels [120], which makes the ecosystem structure of each village different, and then affects the degree of function [121]. The restoration ability of disturbed village ecosystem stability in Karst region and non-Karst region is quite different [122]. The influence of village ecosystem structure on stability is heterogeneous, so it is very important to construct an evaluation index system for the stability of Karst village ecosystem. Due to various natural factors, people will also take different measures to intervene for survival, resulting in obvious differences in livelihood structure [109]. When the natural conditions of the region are difficult, people will go out to work, and then the shortage of village labor force will change the population structure, which will cause disturbance to the stability of the ecosystem. Therefore, based on the concept of stability, ecosystem disturbance intensity was introduced, and the stability evaluation index system was established from three aspects: resistance stability, resilience stability and ecosystem disturbance intensity. According to the characteristics of the structure and stability of the village ecosystem and its mechanism, the stability evaluation index system should be constructed for the KDC village, which will provide theoretical support for the stability improvement strategy of the

KDC village ecosystem, and provides scientific reference for the prevention and control of desertification in villages.

### 5.1.5. In View of the Unclear Mechanism between the Structure and Stability of KDC Village Ecosystem, It Is Necessary to Clarify the Adjustment Mechanism of Structure–Function–Stability through Qualitative and Quantitative Analysis

Studies have shown that the village ecosystem results from long-term co-evolution and development of human, biological and village environments [123]. It is a relatively stable ecosystem with coordinated internal structure and function, and the structure of the village ecosystem determines its function [82]. In order to clarify the mechanism of structure–function–stability scientifically and accurately and to meet the livelihood objectives of farmers and the needs of ES, the mechanism was quantitatively expounded through evaluation indexes [124]. When the village ecosystem in the Karst area is affected by the unreasonable use of human activities, it is easy to cause the ecological environment problem of Karst desertification, which threatens the sustainable development of Karst villages [125]. The KDC village structure has strong vulnerability and is easily affected by different interference factors. Therefore, through qualitative analysis to explore the function-based structural features, transforming ecosystem functions to landscape level and connecting with human needs is the primary problem facing the application of ES in landscape at present [126]. In addition, due to the fragile geographical environment of KDC villages, when the material provided by the natural ecological environment cannot meet the living needs of local villagers, people will choose to go out for work to balance the environment and human needs, reduce environmental pressure and increase economic income. It also changes the population and livelihood structure of villages. Through qualitative analysis, it is clarified that the diversity of livelihood structure in the village ecosystem structure can improve the living feature and promote the stability of the village ecosystem. Therefore, in order to reasonably analyze the mechanism of structure–function–stability interaction of ecosystem and its feedback regulation mechanism, qualitative and quantitative evaluation should be combined.

### 5.1.6. In View of the One-Sidedness of the Research on the Structure of Karst Village Ecosystem, Attention Should Be Paid to the Regulating Role of Human in the Village Ecosystem

Currently, most of the studies on village ecosystem from the aspects of structure and biodiversity to explore the function and supply capacity of the ecosystem, so as to optimize the feature and promote the adjustment of the structure. In the ecosystem, producers, consumers and decomposer make the system function properly through the circulation of matter and the flow of energy. However, in the village ecosystem, farmers are the participants and basic decision-makers of the landscape pattern and production activities in village areas, so the research on village issues should focus on the livelihood of farmers [95]. In the Karst region, population distribution was the biggest threat to ES [127]. As a regulator, human beings should play a non-negligible role in the village ecosystem [128]. This effect may be positive or negative. Human interference in villages is unpredictable. With the change of the way and degree of human interference, the ecosystem succession of villages will accelerate, delay or even change the process. The relationship between human disturbance and ecological succession in village ecosystem is an important research content in village ecology [82]. Therefore, we believe that focusing on human regulation is an indispensable part of the village ecosystem.

### 5.2. Enlightenment on the KDC Villages

#### 5.2.1. Focus on Ecosystem Structural Adjustment to Enhance Stability

Structure is a key feature of the village ecosystem. The function of any ecosystem is based on its unique structure. Irrational human exploitation and use has led to the destruction of the structure and function of fragile ecosystems [129]. When the ecosystem structure is complex, its function and habitat conditions are good, and likewise the more

resistant to disturbance and the more stable it is (Figure 5) [130]. The complexity of ecosystem structure is lower in Wangjiazhai village (A) than in Youqi village (B), which shows a diversity of arable land characteristics and a high accessibility of roads as corridors, but not in Wangjiazhai. Therefore, the support services, supply services and cultural services in Wangjiazhai Village are lower than those in Youqi Village. However, because of the wide forest area in Wangjiazhai, its regulating services are better than other services. Therefore, KDC villages shall emphasize the restructuring of ecosystems, improve the heterogeneity of landscape patterns and enhance the stability of ecosystems.

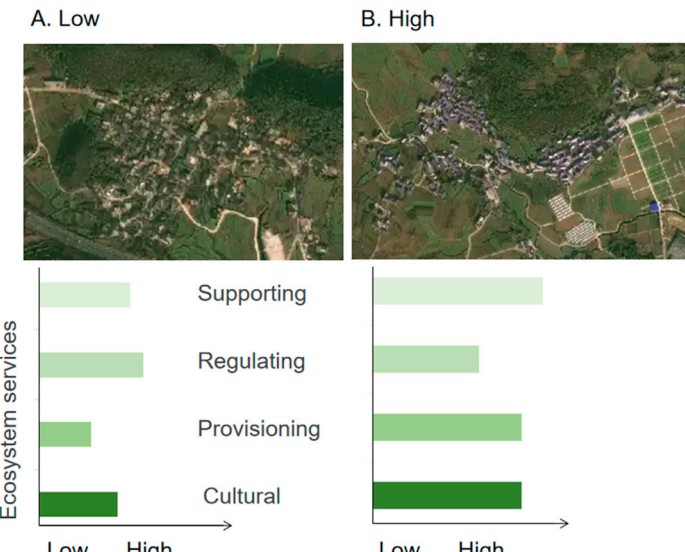

**Figure 5.** Diversity degree of landscape pattern and its service ability.

5.2.2. The Evaluation Models and Index Systems of Structure and Stability of Village Ecosystem Should Be Constructed, Which Is the Key to Correctly Evaluating Structure and Stability

The improvement or degradation of ecosystem in KDC areas directly affects its service capacity [131]. Correctly evaluating the structure of the village ecosystem and accurately establishing the index system of the stability can combine the qualitative and quantitative studies and increase their scientific nature. Therefore, the evaluation model and index system should be established scientifically, according to the structural, landscape ecological and socio-economic characteristics of the village ecosystem in the KDC area.

5.2.3. The Analysis of Village Ecosystem Characteristics Should Be Strengthened, Which Is an Important Way to Clarify the Mechanisms of Stability

Village ecosystem structure optimization is the foundation of the ecological system stability. The analysis of village ecosystem characteristics can be used to clarify the mechanism of improving the system stability. Specifically, on the basis of indicating the relationship between the ecosystem structure of villages in Karst desertification environment and the stability of villages under KDC, the characteristics of village ecosystem structure are identified and analyzed, and the interaction of different structures and their contributions to the stability changes are decomposed. Promote village ecosystem, production, living, such as system service ability, enhance the system stability [106,107]. By analyzing the structural characteristics of the village ecosystem, the structure configuration of the ecosystem is adjusted and optimized, and the formation mechanism of the ecosystem stability will be clarified, which provides a theoretical support for decision-makers to formulate measures to control desertification.

5.2.4. The Regulation of the Human Role in Karst Ecosystem Village Research Should Be Strengthened

Human production and development depend on the natural surroundings, and the appearance of humans means that the natural circumstances have entered another stage of qualitative change. Villages are places where human beings live and settle. In the KDC area, with the continuous increase of population, blind activities have caused many ecological problems that are not conducive to human survival and development. The village structure is unbalanced, and the stability of the village ecosystem is threatened. With the decline of the service capacity of the system, human beings began to reflect, and the deterioration of the ecological circumstances of KDC areas, to protect the Karst area beautiful environment heritage sites [132]. Therefore, the study on the regulating effect of people in the KDC village can promote the ecological restoration and economic development of the KDC village.

## 6. Conclusions

Literature search and screening were conducted on the structure and stability of the village ecosystem in two databases of Web of Science and CNKI, and 105 papers were systematically analyzed and reviewed. The main conclusions are as follows: (1) The number of studies on the KDC village ecosystem shows a significant increasing trend, which indicates that the research on the structure and stability of the village ecosystem has a broad prospect. (2) According to the structural characteristics of the KDC village ecosystem, the structural evaluation index system can be established to optimize the structure. (3) The influence mechanism of the internal and external structure of the village ecosystem on the stability of the village ecosystem is expounded, and the strategies to improve the stability of the village ecosystem are put forward. It is very important to keep long-term effective and sufficient service supply capacity in the process of KDC.

Therefore, when formulating relevant policies, decision-makers should adapt to the relationship between structure and stability, adjust the internal and external structure of village ecosystem, integrate the relationship between rivers, roads and farmland, optimize the farming system, etc. Only by developing village economy and ecology together can sustainable development of villages be promoted.

Achievements were summarized from the structural characteristics, structural optimization, structure and function, stability and other aspects of the village ecosystem. Key scientific questions that need to be addressed within the scope of this study are also explored, indicating research directions for the optimization and stability improvement of the village ecosystem structure in the future. The study can be used for policy optimization and land management by decision-makers to improve village ecosystem services and promote village revitalization and economic development in Karst areas.

**Author Contributions:** K.X. conceived the framework, secured funding and oversaw the entire project; L.L., Q.W., R.Z. and J.Z. collected data; L.L. analyzed the data and wrote the manuscript; K.X. provided comments; K.X. and L.L. reviewed the final manuscript. All authors have read and agreed to the published version of the manuscript.

**Funding:** Funding for the project was supported by the Philosophy and Social Science Planning Key Project of Guizhou Province, China (21GZZB43), the Key Science and Technology Program of Guizhou Province (No. 5411 2017 QKHPTRC) and the China Overseas Expertise Introduction Program for Discipline Innovation (D17016).

**Data Availability Statement:** Not applicable.

**Acknowledgments:** Not applicable.

**Conflicts of Interest:** The authors declare no conflict of interest.

## Abbreviations

KDC      Karst Desertification Control
ES       Ecosystem Services
CNKI     China National Knowledge Network
WOS      Web of Science
PLE      Production–Living–Ecology

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
