# Peer review of "A Review of Village Ecosystem Structure and Stability: Implications for the Karst Desertification Control"

_land, doi:10.3390/land12061136_

Round 1

Reviewer 1 Report

1.      Perhaps due to the problem of sample size, this paper did not use the commonly used knowledge measurement software, such as Citespace,Bibexcel and Vosviewer, and only carried out simple statistical analysis, with low credibility. The above software is recommended in order to achieve more scientific and credible results.

2.      The choice of subject words or keywords can affect the results of the literature. It is suggested to add "rurale cosystem health; spatiotemporal pattern ecosystem services

3.      ROW 21-31, The questions raised in the abstract do not match the focus of future research.

4.      ROW 588-560, According to the content of the article(row450-454), only the content of the evaluation index system is put forward, and the index system is not constructed

5.      ROW783, Your writing style is similar to the structure of this published paper, You can refer to it, but please be aware of academic misconduct.

Suggest appropriate modifications

Author Response

Dear Reviewers,

Thank you very much for your comprehensive and constructive comments on our article, which were very helpful for us to improve this manuscript. This article has been carefully revised with the joint efforts of all authors.

We have carefully considered all comments from the reviewers and revised our manuscript accordingly. The manuscript has also been double-checked, and the typos and grammar errors we found have been corrected. In the following section, we summarize our responses to each comment from the reviewers. We believe that our responses have well addressed all concerns from the reviewers. We hope our revised manuscript can be accepted for publication. The specific modifications are as follows:

Point 1: Perhaps due to the problem of sample size, this paper did not use the commonly used knowledge measurement software, such as Citespace,Bibexcel and Vosviewer, and only carried out simple statistical analysis, with low credibility. The above software is recommended in order to achieve more scientific and credible results.

Response 1: Thank you for your professional and meaningful advice. We have thought about this seriously, but for the following reasons, we still want to keep the manuscript as it is, and we hope that you can recognize our decision.

First of all, these software is very friendly for literature analysis with a large sample size. However, since the content we want to study belongs to interdisciplinary, comprehensive and novel research, we must manually eliminate the literature that has some relevance but is not within our research scope, resulting in a small number of literature conforming to the research scope.

Secondly, some software can not extract information according to our analysis requirements.Based on our research scope and research objectives, the use of these software for analysis neither plays into the advantages of the software nor meets our analysis requirements.

Therefore, we did not consider using these software, but used Excel for manual processing and visual analysis of the selected literature.

Point 2: The choice of subject words or keywords can affect the results of the literature. It is suggested to add "rural ecosystem health; spatiotemporal pattern ecosystem services.

Response 2: Thank you for your professional and meaningful advice. According to your suggestion, we have read and understood the articles related to ecosystem health. We have learned that the content of ecosystem health is broader than the research topic of this paper. For example, Peng Jian mentioned in his article "New research progress and trends in ecosystem health" that "the health of ecosystem is closer to the interaction of resource environment and social culture". Among them, social culture is beyond the scope of "ecosystem structure and stability".

Secondly, only by clarifying the boundary between structure and stability can we clarify the research content. This paper takes the structure and stability of the village ecosystem as the key words, and the ecosystem services related to the retrieved articles are the ecosystem services determined by the structure. If ecosystem services are taken as the subject term, the search for ecosystem services will not only be about the services of the structure, beyond the boundaries of the study in this paper.

Therefore, we will continue to maintain our research boundaries and not expand outward. 

Point 3: ROW 21-31, The questions raised in the abstract do not match the focus of future research.

Response 3: Thank you for your comments. Through a thorough reading of the paper, we have modified the questions raised in the abstract to conform to the future research focus of the paper. See ROW 23-29 for details:

【Before revision】

...Therefore, future research should pay attention to the following three aspects: to deepen the concepts and characteristics of structure and stability; to establish an appropriate village ecosystem structure based on the characteristics of Karst Desertification Control (KDC) village structures; The research on the mechanism of structural influence on stability needs to be strengthened in order to lay a solid foundation for the proposed strategy of stability of village ecosystems in KDC. By applying the strategy of structure optimization and stability improvement to KDC village ecosystem, the service function of karst village ecosystem can be improved, which can provide scientific reference for the sustainable development of KDC village ecosystem...

【After revision】

...Therefore, future research should pay attention to the following three aspects: to strengthen the differentiation research on the spatio-temporal scale, qualitative and quantitative analysis of the influence of Karst Desertification Control (KDC) village structure on the stability; based on the mechanism of structure on function, appropriate village ecosystem structure should be established to improve ecosystem service function; based on the influence mechanism of structure on stability, the stability evaluation index system will be constructed, so as to lay a solid foundation for the stability strategy of KDC village ecosystem...

Point 4: ROW 588-560, According to the content of the article(row450-454), only the content of the evaluation index system is put forward, and the index system is not constructed.

Response 4: Thank you for your comment. According to the questions you raised, we have revised the original manuscript to keep the article consistent. See ROW451-456 and ROW562-563 for details:

ROW450-455:

【Before revision】

...According to the characteristics of the structure and stability of the village ecosystem and its mechanism, the stability evaluation index system is constructed for the KDC village, which provides theoretical support for the stability improvement strategy of the KDC village ecosystem, and provides scientific reference for the prevention and control of desertification in villages...

【After revision】

...According to the characteristics of the structure and stability of the village ecosystem and its mechanism, the stability evaluation index system should be constructed for the KDC village, which will provide theoretical support for the stability improvement strategy of the KDC village ecosystem, and provides scientific reference for the prevention and control of desertification in villages....

ROW561-562:

【Before revision】

...According to the structural characteristics of KDC village ecosystem, a unified structural optimization model is established...

【After revision】

...According to the structural characteristics of KDC village ecosystem, the structural evaluation index system can be established to optimize the structure...

Point 5: ROW783, Your writing style is similar to the structure of this published paper, You can refer to it, but please be aware of academic misconduct.

Response 5: First of all, I would like to thank the reviewers for their reminding.

Secondly, the article mentioned by ROW787-798 is “Research Progress of Grassland Ecosystem Structure and Stability and Inspiration for Improving Its Service Capacity in the Karst Desertification Control”. In that paper, the structure and stability of grassland ecosystem are discussed.

We takes the village ecosystem as the research object to discuss the influence of structure on stability in the village ecosystem. In terms of structure, this paper draws on the good points of that article under the premise of retaining its own characteristics. In terms of content, this paper focuses on the study of villages, so there is no academic misconduct.

Of course, in future studies, we will bear in mind the reviewer's comments and avoid academic misconduct.

We have tried our best to make our statements clearer and more concise throughout the text.  We hope that this revision will be approved.

On behalf of our team, I would like to thank you for your great efforts in guiding our research and for your kindness.

Reviewer 2 Report

Dear Authors

You have prepared a very good manuscript, congratulation. Just a few questions for me as a reader:

1- Why other databases have not been investigated?

2- Can the considered databases provide full coverage of the topic under review?

3- Be sure to mention that in different countries, articles on your research may have been published in the official language of the country and in local journals.

Thanks

Minor editing of English language required

Author Response

Dear Reviewers,

Thank you very much for your comprehensive and constructive comments on our article, which were very helpful for us to improve this manuscript. This article has been carefully revised with the joint efforts of all authors.

We have carefully considered all comments from the reviewers and revised our manuscript accordingly. The manuscript has also been double-checked, and the typos and grammar errors we found have been corrected. In the following section, we summarize our responses to each comment from the reviewers. We believe that our responses have well addressed all concerns from the reviewers. We hope our revised manuscript can be accepted for publication. The specific modifications are as follows:

Point 1: Why other databases have not been investigated?

Response 1: Thank you for your comments. First of all, from the perspective of database reliability, Web of Science is an important database for obtaining global academic information. It has collected more than 20,000 world authoritative and high-impact academic journals and conference documents. CNKI is the most authoritative database in China at present. It has a powerful network database full-text retrieval system with large information content, wide coverage, rapid and timely update, complete retrieval service functions, and the quality of its literature can be fully guaranteed.

Secondly, the two databases of WOS and CNKI can complement each other effectively.

Finally, because the school only has access to these two databases, it is easy to search and filter.

Therefore, we reviewed both WOS and CNKI databases.

Point 2: Can the considered databases provide full coverage of the topic under review?

Response 2: Thank you for your valuable comments. From the point of view of the collection range, the data collected by Web of Science and CNKI are large. This paper belongs to the interdisciplinarity of natural science and social science. These two databases can comprehensively contain the topics studied.

The WOS core database covers the fields of natural sciences, engineering technology, biomedicine, social sciences, arts and humanities.

CNKI covers basic science, social science, agriculture, economics and management science, etc.

Point 3: Be sure to mention that in different countries, articles on your research may have been published in the official language of the country and in local journals.

Response 3: Thank you for your comments. As a result of your comments, we have made changes in the article, as detailed in ROW 158-161:

【Before revision】

...According to the number of published literature, the scientific literature research on the structure and stability of village ecosystems covers 32 countries...

【After revision】

...In terms of the number of published literature, the scientific literature research on the structure and stability of village ecosystems covers 32 countries, except for a few articles published in national official languages and local journals that have not been searched...

In addition, it should be noted that at present, most researchers focus on natural ecosystems, and there are few studies on the topic of this paper. There is a small probability that it may be published in a country's official language or local journal. Even if it does, it cannot be read and studied by popular scholars due to access rights and different languages.

Moreover, due to the borderless academic field, all high-quality articles are presented in English to facilitate readers' learning and communication. Therefore, it is necessary to study and publish articles on the structure and stability of village ecosystem, so as to provide a theoretical basis for sustainable village development.

We have tried our best to make our statements clearer and more concise throughout the text.  We hope that this revision will be approved.

On behalf of our team, I would like to thank you for your great efforts in guiding our research and for your kindness.

Author Response

Dear Reviewers,

Thank you very much for your comprehensive and constructive comments on our article, which were very helpful for us to improve this manuscript. This article has been carefully revised with the joint efforts of all authors.

We have carefully considered all comments from the reviewers and revised our manuscript accordingly. The manuscript has also been double-checked, and the typos and grammar errors we found have been corrected. In the following section, we summarize our responses to each comment from the reviewers. We believe that our responses have well addressed all concerns from the reviewers. We hope our revised manuscript can be accepted for publication. The specific modifications are as follows:

Point 1:Abstract:

Line 12-13: I suggest that the authors delete the words “for us” and rewrite the sentence “Therefore, it is necessary to study…”.

Response 1: Thank you for your comments. By reading Line 12-13, we delete the word "for us" and rewrite the sentence as“Therefore, it is of great necessity to study the structure and stability of village ecosystem and optimize the structure of village ecosystem to better guide spatial planning and restore village ecology.”

Point 2: Introduction:

Line 93: change the word “comb” to “search” in this sentence.

Response 2: Thank you for your valuable comments. We have changed "comb" to "search" in this sentence based on your comment, as detailed in lines 94-96:

...Emerge, therefore, the current articles systematically search, in order to better grasp the direction of development, to guide the research on the stability of village ecosystem structure and development...

Point 3: Results:

Line 178 (3.4): The authors should incorporate “and” instead of the symbol (&) and rewrite “Northwest Agriculture and Forestry University”.

Response 3: Thank you for your comments. We have removed the symbol (&) in accordance with your comment, added "and" and rewritten "Northwest Agriculture and Forestry University" as detailed in line 172-173 (3.4) :

...there are Northwest Agriculture and Forestry University...

Point 4: Strength and weakness:

The materials and methods, results, discussion, and conclusions were properly highlighted and explained.

However, to enrich the manuscript, I suggest the authors should include a short paragraph into the conclusion section explaining the “concept of policy recommendations and sustainable environment” as related to this study.

Response 4: Thank you for your comments.We have added a short paragraph to the conclusion explaining the policy recommendations and sustainability, as detailed in lines 567-571:

...Therefore, when formulating relevant policies, decision-makers should adapt to the relationship between structure and stability, adjust the internal and external structure of village ecosystem, integrate the relationship between rivers, roads and farmland, optimize the farming system, etc. Only by developing village economy and ecology together can sustainable development of villages be promoted...

Point 5: References:

Line 632-634: The authors should darken “the year of journal” for #17 in the reference list.

Response 5: Thank you for your comments, we have revised according to your comments, details can be found in the article line 638-640:

Xiao, J.; Xiong , K.N. A review of agroforestry ecosystem services and its enlightenment on the ecosystem improvement of rocky desertification control. Science of the Total Environment 2022, 852(1), 158538. https://doi.org/10.1016/j.scitotenv.2022.158538

Line 678-679: The authors should italicize “the name of journal” and darken “the year of journal” for #35 in the reference list.

Thank you for your comments, we have revised according to your comments, details can be found in the article line 683-684:

Wang, Y.L.; Zhao, Y.B.; Han, D. The spatial structure of landscape eco-systems: concept, indices and case studies. Advances in Earth Science 1999, 14(3), 24-30.

Take Note: Always maintain consistency in writing your references. If you want to use “sentence case” or “capitalize each word”, adopt one and maintain it throughout your list of references.

Line 680-681 (#36): Rewrite this reference to conform with the above stated expectations.

Thank you for your comments. Line 685-686 (#36) is a master's thesis. As required by this journal, "Author 1, A.B. Title of Thesis. Level of Thesis, Degree-Granting University, Location of University, Date of Completion. ". So we're going to keep.

Line 732-734: The authors should darken “the year of journal” for #60 in the reference list.

Thank you for your comments, we have modified according to your comments, details can be found on Line737-738:

Baumgartner, L.J.; Marsden, T.; Duffy, D.; Horta, A; Ning, N. Optimizing efforts to restore aquatic ecosystem connectivity requires thinking beyond large dams. Environmental Research Letters 2022, 17(1), 4008. https://doi.org/10.1088/1748-9326/ac40b0

Line 791-792: The authors should darken “the year of journal” for #84 in the reference list.

Thank you for your comments, we have modified according to your comments, details can be found on Line795-796:

Yang, X.; Wei, Q.L.; Du, C.L.; Wang, Y.D. The Commonness and Differentiation of Influencing Factors of Domestic China's Rural Settlements Morphology in the Past Decade. Planners 2019, 35(18), 19-25.

Line 810-811: The authors should darken “the year of journal” for #92 in the reference list.

Thank you for your comments, we have modified according to your comments, details can be found on Line 814-815:

Su, W.C. A preliminary discussion on the structural model of ecosystem stability in Karst basin. Guizhou Science 2002, 20(01), 14-20.

We have tried our best to make our statements clearer and more concise throughout the text.  We hope that this revision will be approved.

On behalf of our team, I would like to thank you for your great efforts in guiding our research and for your kindness.

Reviewer 4 Report

The topic of the article is interesting, and the results of this study have important significance for improving the service supply capacity of village ecosystem and promoting the control of rocky desertification. The choice of the research problem should be assessed as accurate and justified in both the scientific and practical aspects. Overall, I found a lot of effort the authors made to achieve their goal. I appreciate their process of developing this research article.

The title of the article is corresponding to its content. The issues included in the article can be used in practice or for the development of the theory. The structure of the study is clear, coherent, and consequent. The choice of the literature is correct and current. The study is a contribution to further theoretical considerations and empirical research.

This article has a certain level of readability and is fairly well-written. However, it is suggested that the author(s) improve their use of language to achieve a higher standard of writing in science.

Author Response

Dear Reviewers,

Thank you very much for your comprehensive and constructive comments on our article, which were very helpful for us to improve this manuscript. This article has been carefully revised with the joint efforts of all authors.

We have carefully considered all comments from the reviewers and revised our manuscript accordingly. The manuscript has also been double-checked, and the typos and grammar errors we found have been corrected. In the following section, we summarize our responses to each comment from the reviewers. We believe that our responses have well addressed all concerns from the reviewers. We hope our revised manuscript can be accepted for publication. The specific modifications are as follows:

Point 1: It is suggested that the author(s) improve their use of language to achieve a higher standard of writing in science.

Response 1: Thank you for your comment. By combing through the full text, we have made language modifications to the following points:

ROW 12-14, delete "for us":...Therefore, it is of great necessity to study the structure and stability of village ecosystem and optimize the structure of village ecosystem to better guide spatial planning and restore village ecology...

ROW 15-16:...We reviewed 105 relevant articles...

ROW 42-46:...Some villages used the concept of village ecosystem structure optimization and ecological stability to alleviate the fragmentation of village landscape, coordinate the relationship between village development and ecosystem, and realize the sustainability of village development...

ROW 94-96:

...Emerge, therefore, the current articles systematically search, in order to better grasp the direction of development, to guide the research on the stability of village ecosystem structure and development...

ROW 172-174:...The relevant units are mainly universities: there are Northwest Agriculture and Forestry University, Chongqing University, Beijing Forestry University, Beijing Normal University, Guilin University of Technology, etc...

ROW 192-193:...Changes in ecosystem structure affect the land structure of PLE of village, ES, and farmers' livelihoods...

We have tried our best to make our statements clearer and more concise throughout the text.  We hope that this revision will be approved.

On behalf of our team, I would like to thank you for your great efforts in guiding our research and for your kindness.

Round 2

Reviewer 1 Report

参考的格式,细节需要注意。

例如 row755/804/822